# Adaptive Reconstruction of Imperfectly Observed Monotone Functions, with Applications to Uncertainty Quantification

**Luc Bonnet [1,2]**, **Jean-Luc Akian [1]**, **Éric Savin [1,\*]** and **T. J. Sullivan [3,4]**

[1] ONERA, 29 Avenue de la Division Leclerc, 92320 Châtillon, France; luc.bonnet@onera.fr (L.B.); jean-luc.akian@onera.fr (J.-L.A.)

[2] Laboratoire MSSMat—UMR CNRS 8579, CentraleSupélec, 8–10 rue Joliot Curie, 91190 Gif sur Yvette, France

[3] Mathematics Institute and School of Engineering, University of Warwick, Coventry CV4 7AL, UK; t.j.sullivan@warwick.ac.uk or sullivan@zib.de

[4] Zuse Institute Berlin, Takustraße 7, 14195 Berlin, Germany

[\*] Correspondence: eric.savin@onera.fr

**Abstract:** Motivated by the desire to numerically calculate rigorous upper and lower bounds on deviation probabilities over large classes of probability distributions, we present an adaptive algorithm for the reconstruction of increasing real-valued functions. While this problem is similar to the classical statistical problem of isotonic regression, the optimisation setting alters several characteristics of the problem and opens natural algorithmic possibilities. We present our algorithm, establish sufficient conditions for convergence of the reconstruction to the ground truth, and apply the method to synthetic test cases and a real-world example of uncertainty quantification for aerodynamic design.

**Keywords:** adaptive approximation; isotonic regression; optimisation under uncertainty; uncertainty quantification; aerodynamic design

---

## 1. Introduction

This paper considers the problem of adaptively reconstructing a monotonically increasing function $F^\dagger$ from imperfect pointwise observations of this function. In the statistical literature, the problem of estimating a monotone function is commonly known as *isotonic regression*, and it assumed that the observed data consist of noisy pointwise evaluations of $F^\dagger$. However, we consider this problem under assumptions that differ from the standard formulation, and these differences motivate our algorithmic approach to the problem. To be concrete, our two motivating examples are that

$$F^\dagger(x) := \mathbb{P}_{\Xi \sim \mu}[g(\Xi) \le x] \tag{1}$$

is the cumulative distribution function (CDF) of a known real-valued function $g$ of a random variable $\Xi$ with known distribution $\mu$, or that

$$F^\dagger(x) := \sup_{(g,\mu) \in \mathcal{A}} \mathbb{P}_{\Xi \sim \mu}[g(\Xi) \le x] \tag{2}$$

is the supremum of a family of such CDFs over some class $\mathcal{A}$. We assume that we have access to a numerical optimisation routine that can, for each $x$ and some given numerical parameters $q$ (e.g., the number of iterations or other convergence tolerance parameters), produce a *numerical estimate* or *observation* $G(x, q)$ of $F^\dagger(x)$; furthermore, we assume that $G(x, q) \le F^\dagger(x)$ is always true, i.e.,

the numerical optimisation routine always under-estimates the true optimum value, and that the positive error $F^\dagger(x) - G(x,q)$ can be controlled to some extent through the choice of the optimisation parameters $q$, but remains essentially influenced by randomness in the optimisation algorithm for each $x$. The assumption $G(x,q) \leq F^\dagger(x)$ is for example coherent with either Equation (1), which may be approached by increasing the number of samples (say $q$) in a Monte Carlo simulation, or Equation (2), which is a supremum over a set that may be explored only partially by the algorithm.

A single observation $G(x,q)$ yields some limited information about $F^\dagger(x)$; a key limitation is that one may not even know a priori how accurate $G(x,q)$ is. Naturally, one may repeatedly evaluate $G$ at $x$, perhaps with different values of the optimisation parameters $q$, in order to more accurately estimate $F^\dagger(x)$. However, a key observation is that a *suite* of observations $G(x_i, q_i)$, $i = 1, \ldots, I$, contains much more information than simply estimates of $F^\dagger(x_i)$, $i = 1, \ldots, I$, and this information can and must be used. For example, suppose that the values $(G(x_i, q_i))_{i=1}^{I}$ are not increasing, e.g., because

$$G(x_i, q_i) > G(x_{i'}, q_{i'}) \quad \text{and} \quad x_i < x_{i'}.$$

Such a suite of observations would be inconsistent with the axiomatic requirement that $F^\dagger$ is an increasing function. In particular, while the observation at $x_i$ may be relatively good or bad on its own merits, the observation $G(x_{i'}, q_{i'})$ at $x_{i'}$, which violates monotonicity, is in some sense "useless" as it gives no better lower bound on $F^\dagger(x_{i'})$ than the observation at $x_i$ does. The observation at $x_{i'}$ is thus a good candidate for repetition with more stringent optimisation parameters $q$—and this is not something that could have been known without comparing it to the rest of the data set.

The purpose of this article is to leverage this and similar observations to define an algorithm for the reconstruction of the function $F^\dagger$, repeating old observations of insufficient quality and introducing new ones as necessary. The principal parameter in the algorithm is an "exchange rate" $\mathcal{E}$ that quantifies the degree to which the algorithm prefers to have a few high-quality evaluations versus many poor-quality evaluations. Our approach is slightly different from classical isotonic (or monotonic) regression, which is understood as the least-squares fitting of an increasing function to a set of points in the plane. The latter problem is uniquely solvable and its solution can be constructed by the pool adjacent violators algorithm (PAVA) extensively studied in Barlow et al. [1]. This algorithm consists of exploring the data set from left to right until the monotonicity condition is violated, and replacing the corresponding observations by their average while back-averaging to the left if needed to maintain monotonicity. Extensions to the PAVA have been developed by de Leeuw et al. [2] to consider non least-squares loss functions and repeated observations, by Tibshirani et al. [3] to consider "nearly isotonic" or "nearly convex" fits, and by Jordan et al. [4] to consider general loss functions and partially ordered data sets. Useful references on isotonic regression also include Robertson et al. [5] and Groeneboom and Jongbloed [6].

The remainder of this paper is structured as follows. Section 2 presents the problem description and notation, after which the proposed adaptive algorithm for the reconstruction of $F^\dagger$ is presented in Section 3. We demonstrate the convergence properties of the algorithm in Section 3.2 and study its performance on several analytically tractable test cases in Section 4. Section 5 details the application of the algorithm to a challenging problem of the form Equation (2) drawn from aerodynamic design. Some closing remarks are given in Section 6.

## 2. Notation and Problem Description

In the following, the "ground truth" response function that we wish to reconstruct is denoted $F^\dagger \colon [a,b] \to \mathbb{R}$ and has inputs $x \in [a,b] \subset \mathbb{R}$. It is assumed that $F^\dagger$ is monotonically increasing and non-constant on $[a,b]$. In contrast, $G \colon [a,b] \times \mathbb{R}_+ \to \mathbb{R}$ denotes the numerical process used to obtain an imperfect pointwise observation $y$ of $F^\dagger(x)$ at some point $x \in [a,b]$ for some numerical parameter $q \in \mathbb{R}_+$. Here, on a heuristic level, $q > 0$ stands for the "quality" of the noisy evaluation $G(x,q)$.

The main aim of this paper is to show the effectiveness of the proposed algorithm for the adaptive reconstruction of $F^\dagger$, which could be continuous or not, from imperfect pointwise observations $G(x_i, q_i)$ of $F^\dagger$, where we are free to choose $x_{i+1}$ and $q_{i+1}$ adaptively-based on $x_j$, $q_j$, and $G(x_j, q_j)$ for $j \leq i$.

First, we associate with $I$ imperfect pointwise observations $\{x_i, y_i := G(x_i, q_i)\}_{i=1}^{I} \subset [a, b] \times \mathbb{R}$, positive numbers $\{q_i\}_{i=1}^{I} \subset \mathbb{R}_+$ which we will call *qualities*. The quality $q_i$ quantifies the confidence we have in the pointwise observation $y_i$ of $F^\dagger(x_i)$ using the numerical process $G(x_i, q_i)$. The higher this value, the greater the confidence. We divide this quality as the product of two different numbers $c_i$ and $r_i$, $q_i = c_i \times r_i$, with the following definitions:

- *Consistency $c_i \in \{0, 1\}$*: This describes the fact that two successive points must be monotonically consistent with respect to each other. That is, when one takes two input values $x_2 > x_1$, one should have $y_2 \geq y_1$ as $y$ must be monotonically increasing. There is no consistency associated with the very first data point as it does not have any predecessor.
- *Reliability $r_i \in \mathbb{R}_+$*: This describes how confident we are about the numerical value. Typically, it will be related to some error estimator if one is available, or the choice of optimisation parameters. It is expected that the higher the reliability, the closer the pointwise observation is to the true value, on average.

Typically, if the observation $y_{i+1} = G(x_{i+1}, q_{i+1})$ is consistent with regard to the observation $y_i = G(x_i, q_i)$ where $x_{i+1} > x_i$, the quality $q_{i+1}$ associated with $y_{i+1}$ will be equal to $q_{i+1} = r_{i+1} \in \mathbb{R}_+^*$ since $c_{i+1} = 1$ in this case. If the value is not consistent, we have $q_{i+1} = r_{i+1} \times c_{i+1} = 0$. Finally, if $x = a$ there is no notion of consistency as there is no point preceding it. Thereby, the quality associated with this point is only equal to its reliability.

Moreover, we associate to these pointwise observations a notion of area, illustrated in Figure 1 and defined as follows. Consider two consecutive points $x_i$ and $x_{i+1}$ with their respective observations $y_i$ and $y_{i+1}$, the area $a_i$ for these two points is

$$a_i = (x_{i+1} - x_i) \times (y_{i+1} - y_i). \tag{3}$$

Thus, we can define a vector $\boldsymbol{a} = \{a_i\}_{i=1}^{I-1}$ which contains all the computed areas for the whole dataset. In addition, we can assure that if we take two points $x_1$ and $x_2 > x_1$ with $y_1 = F^\dagger(x_1)$ and $y_2 = F^\dagger(x_2)$—namely the error at these point is equal to zero, the graph of ground truth function $F^\dagger$ must lie in the rectangular area spanned by the two points $(x_1, F^\dagger(x_1))$ and $(x_2, F^\dagger(x_2))$.

To adopt a conservative point of view, we choose as the approximating function $F$ of $F^\dagger$ a piecewise constant interpolation function, say:

$$F(x) = \sum_{i=1}^{I-1} y_i \mathbb{1}_{[x_i, x_{i+1})}(x), \tag{4}$$

where $\mathbb{1}_{\mathcal{I}}$ denotes the indicator function of the interval $\mathcal{I}$. We do not want this interpolation function to overestimate the true function $F^\dagger$ as one knows that the numerical estimate in our case always underestimates the ground truth function $F^\dagger(x)$. See Figure 1 for an illustration of this choice, which can be viewed as a worst-case approach. Indeed, this chosen interpolation function is the worst possible function underestimating $F^\dagger$ given two points $x_1$ and $x_2$ and their respective observations $y_1$ and $y_2$.

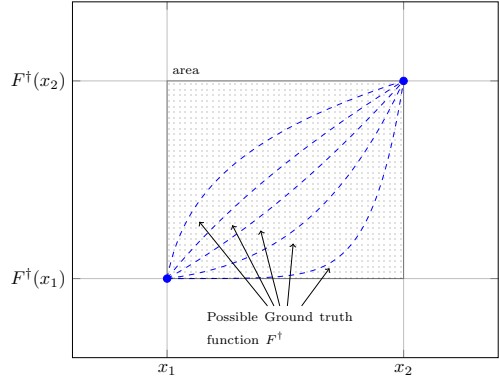
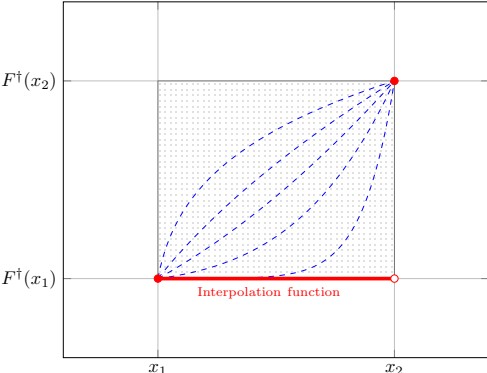

(**a**) Possible ground truth functions between two consecutive points $x_1$ and $x_2$. The ground truth function must lie in the area formed by these two points.

(**b**) Right-continuous piecewise constant interpolation function.

**Figure 1.** Possible ground truth functions between two consecutive points $x_1$ and $x_2$, and our choice of piecewise constant interpolant.

## 3. Reconstruction Algorithms

The reconstruction algorithm that we propose, Algorithm 1, is driven to produce a sequences of reconstructions that converges to $F^\dagger$ by following a principle of *area minimisation*: we associate to the discrete data set $\{x_i, y_i\}_{i=1}^{I} \subset [a, b] \times \mathbb{R}$ a natural notion of area (3) as explained above, and seek to drive this area towards zero. The motivation behind this objective is in Proposition 2 which states that the area converges to 0 as more points are added to the data set. However, the objective of minimising the area is complicated by the fact that evaluations of $F^\dagger$ are imperfect. Therefore, a key user-defined parameter in the algorithm is $\mathcal{E} \in (0, \infty)$, which can be thought of as an "exchange rate" that quantifies to what extent the algorithm prefers to redo poor-quality evaluations of the target function versus driving the area measure to zero.

### 3.1. Algorithm

The main algorithm is organized as follows, starting from $I^{(0)} \geq 2$ points and a dataset that is assumed to be consistent at the initial step $n = 0$. It goes through $N$ iterations, where $N$ is either fixed a priori, or obtained a posteriori once a stopping criterion is met. Note that $q_{\text{new}}$ stands for the quality of a newly generated observation $y_{\text{new}}$ for any new point $x_{\text{new}}$ introduced by the algorithm. The latter is driven by the user-defined "exchange rate" $\mathcal{E}$ as explained just above. At each step $n$, the algorithm computes the weighted area $\text{WA}^{(n)}$ as the minimum of the quality times the sum of the areas of the data points:

$$\text{WA}^{(n)} = q_-^{(n)} \times \text{A}^{(n)}, \tag{5}$$

where

$$q_-^{(n)} = \min_{1 \leq i \leq I^{(n)}} \{q_i^{(n)}\}, \quad \text{A}^{(n)} = \sum_{i=1}^{I^{(n)}-1} a_i^{(n)}, \tag{6}$$

$a_i^{(n)}$ is the area computed by Equation (3) at step $n$ (see also Equation (9)), and $I^{(n)}$ is the number of data points. Then it is divided into two parts according to the value of $\text{WA}^{(n)}$ compared to $\mathcal{E}$.

---

**Algorithm 1:** Adaptive algorithm to reconstruct a monotonically increasing function $F$[†]

---

**Input:** $I^{(0)} \geq 2$, $\{x_i^{(0)}, y_i^{(0)}, q_i^{(0)}\}_{i=1}^{I^{(0)}}$ and $\mathcal{E}$.

**Output:** $\{x_i^{(N)}, y_i^{(N)}, q_i^{(N)}\}_{i=1}^{I^{(N)}}$ with $I^{(N)} \geq I^{(0)}$.

**Initialization**:

Get the worst quality point and its index:

- $q_-^{(0)} = \min\limits_{1 \leq i \leq I^{(0)}} \{q_i^{(0)}\}$;
- $i_-^{(0)} = \arg\min\limits_{1 \leq i \leq I^{(0)}} \{q_i^{(0)}\}$.

Compute the area of each pair of data points: $a_i^{(0)} = (x_{i+1}^{(0)} - x_i^{(0)}) \times (y_{i+1}^{(0)} - y_i^{(0)})$.

Get the biggest rectangle and its index:

- $a_+^{(0)} = \max\limits_{1 \leq i \leq I^{(0)}-1} \{a_i^{(0)}\}$;
- $i_+^{(0)} = \arg\max\limits_{1 \leq i \leq I^{(0)}-1} \{a_i^{(0)}\}$.

Define the weighted area at step $n = 0$ as $\mathrm{WA}^{(0)} = q_-^{(0)} \times \sum\limits_{i=1}^{I^{(0)}-1} a_i^{(0)}$.

**while** $n \leq N$ **do**

    **if** $\mathrm{WA}^{(n)} < \mathcal{E}$ **then**

        Data points are unchanged: $I^{(n+1)} = I^{(n)}$ and $\{x_i^{(n+1)}\}_{i=1}^{I^{(n+1)}} = \{x_i^{(n)}\}_{i=1}^{I^{(n)}}$;

        Store the old value $y_{\mathrm{old}} = y_{i_-^{(n)}}^{(n)}$;

        **while** $y_{\mathrm{new}} \leq y_{\mathrm{old}}$ **do**

            Compute a new value $y_{\mathrm{new}} = G(x_{i_-^{(n)}}^{(n)}, q_{\mathrm{new}})$;

        **end**

    **else**

        Introduce a new point at the middle of the biggest rectangle: $I^{(n+1)} = I^{(n)} + 1$,

        $x_{\mathrm{new}} = \frac{1}{2}(x_{i_+^{(n)}}^{(n)} + x_{i_+^{(n)}+1}^{(n)})$, and

        $(x_1^{(n+1)}, \ldots, x_{i_+^{(n)}}^{(n+1)}, x_{i_+^{(n)}+1}^{(n+1)}, x_{i_+^{(n)}+2}^{(n+1)}, \ldots, x_{I^{(n+1)}}^{(n+1)}) = (x_1^{(n)}, \ldots, x_{i_+^{(n)}}^{(n)}, x_{\mathrm{new}}, x_{i_+^{(n)}+1}^{(n)}, \ldots, x_{I^{(n)}}^{(n)})$;

        Compute the new value $y_{\mathrm{new}} = G(x_{\mathrm{new}}, q_{\mathrm{new}})$;

    **end**

    Verify consistency of the pointwise observations $\{y_i^{(n+1)})\}_{i=1}^{I^{(n+1)}}$ by checking their quality. If there are not consistent, recompute them until obtaining consistency and then update the quality vector;

    Compute the new quality vector $\{q_i^{(n+1)}\}_{i=1}^{I^{(n+1)}}$ and area vector $\{a_i^{(n+1)}\}_{i=1}^{I^{(n+1)}}$;

    Update $q_-^{(n+1)}, i_-^{(n+1)}, a_+^{(n+1)}$ and $i_+^{(n+1)}$;

    Compute $\mathrm{WA}^{(n+1)} = q_-^{(n+1)} \times \sum\limits_{i=1}^{I^{(n+1)}-1} a_i^{(n+1)}$;

    $n = n + 1$;

**end**

---

- If $\mathrm{WA}^{(n)} < \mathcal{E}$, then the algorithm aims at increasing the quality $q_-^{(n)}$ of the worst data point (the one with the lowest quality) with index $i_-^{(n)} = \arg\min_{1 \leq i \leq I^{(n)}} \{q_i^{(n)}\}$ at step $n$. It stores the corresponding old value $y_{\mathrm{old}}$, searches for a new value $y_{\mathrm{new}}$ by improving successively the quality of this very point, and stops when $y_{\mathrm{new}} > y_{\mathrm{old}}$.
- If $\mathrm{WA}^{(n)} \geq \mathcal{E}$, then the algorithm aims at driving the total area $\mathrm{A}^{(n)}$ to zero. In that respect, it identifies the biggest rectangle

$$a_+^{(n)} = \max_{1 \leq i \leq I^{(n)}-1} \{a_i^{(n)}\} \tag{7}$$

and its index

$$i_+^{(n)} = \underset{1 \le i \le I^{(n)}-1}{\arg\max} \{a_i^{(n)}\} \tag{8}$$

and adds a new point $x_{\text{new}}$ at the middle of this biggest rectangle. Then, it computes a new data value $y_{\text{new}} = G(x_{\text{new}}, q_{\text{new}})$ with a new quality $q_{\text{new}}$.

In both cases, the numerical parameters $q_{\text{new}}$ (for example several iterations, or the size of a sampling set or a population) are arbitrary and any value can be chosen in practice each time a new point $x_{\text{new}}$ is added to the dataset. They can be increased arbitrarily as well each time such a new point has to be improved. Indeed, the numerical parameters $q$ of the optimisation routine we have access to can be increased as much as desired, and increasing them will improve the estimates $G(x, q)$ of the true function $F^\dagger(x)$ uniformly in $x$; see Assumption 1. The algorithm then verifies the consistency of the dataset by checking the quality of each point. If there is any inconsistent point, the algorithm computes a new value until obtaining consistency by improving successively the corresponding reliability. This is achieved in a finite number of steps starting from an inconsistent point and exploring the dataset from the left to the right.

Finally, the algorithm updates the quality vector $\{q_i^{(n+1)}\}_{i=1}^{I^{(n+1)}}$, the area vector $\{a_i^{(n+1)}\}_{i=1}^{I^{(n+1)}}$, the worst quality $q_-^{(n+1)}$ and the index $i_-^{(n+1)}$ of the corresponding point, the biggest rectangle $a_+^{(n+1)}$ and its index $i_+^{(n+1)}$, and then the new weighted area $\text{WA}^{(n+1)}$.

### 3.2. Proof of Convergence

We denote by $I^{(n)}$ the number of data points, and $\{x_i^{(n)}, y_i^{(n)}, q_i^{(n)}\}_{i=1}^{I^{(n)}}$ the positions of the data points, the observations given by the optimization algorithm at these positions, and the qualities associated with the optimization algorithm at the step $n$ of Algorithm 1. For each $i = 1, \ldots, I^{(n)} - 1$, we define $s_i^{(n)} = [x_i^{(n)}, x_{i+1}^{(n)}[ \subset [a, b]$ and the vector containing all rectangle areas $\{a_i^{(n)}\}_{i=1}^{I^{(n)}-1}$ by:

$$a_i^{(n)} = (x_{i+1}^{(n)} - x_i^{(n)}) \times (y_{i+1}^{(n)} - y_i^{(n)}). \tag{9}$$

The pointwise observation $y_i^{(n)} = G(x_i^{(n)}, q_i^{(n)})$ is thus associated with the quality $q_i^{(n)} \in \mathbb{R}_+$, which quantifies the confidence we have in this observation as outlined in the problem description in Section 2. This number can represent the inverse error achieved by the optimization algorithm, for example, or the number of iterations, or the number of individuals in a population, or any other numerical parameter pertaining to this optimization process. The higher it is, the closer the observation is to the true target value. Therefore we consider the following assumption on the numerical process $G$.

**Assumption 1.** *$G(x, q)$ converges to $F^\dagger(x)$ as $q \to +\infty$ uniformly in $x \in [a, b]$, that is:*

$$\forall \epsilon > 0, \ \exists Q > 0 \text{ such that } \forall q \ge Q, \ \forall x \in [a, b], \ \left| G(x, q) - F^\dagger(x) \right| \le \epsilon.$$

Moreover, we can guarantee that:

$$\forall x \in [a, b], \quad \forall q \in \mathbb{R}_+, \quad G(x, q) \le F^\dagger(x). \tag{10}$$

That is, the optimisation algorithm will always underestimate the true value $F^\dagger(x)$. In this way, one can model the relationship between the numerical estimate $G$ and the true value $F^\dagger$ as:

$$\forall x \in [a, b], \quad \forall q \in \mathbb{R}_+, \quad G(x, q) = F^\dagger(x) - \epsilon(x, q), \tag{11}$$

where $\epsilon$ is a positive random variable. These assumptions imply some robustness and stability of the algorithm we use.

In the following, we will assume that $I^{(0)} \ge 2$. That is, we have at least two data points at the beginning of the reconstruction algorithm. Also among these points, we have one point at $x = a$ and

another one at $x = b$. Moreover, we will assume that the initial dataset is consistent. Since Algorithm 1 recomputes the inconsistent points at all steps, we can also consider in the following that any new numerical observation is actually consistent. Also, we need to guarantee that the weighted area $\text{WA}^{(n)}$ will permanently oscillate about $\mathcal{E}$ as the iteration step $n$ is increasing; this is the purpose of Assumption 3 below as shown in the subsequent Proposition 1. From these properties it will then be shown that Algorithm 1 is convergent, as stated in Theorem 1.

**Assumption 2.** *Any new numerical value obtained by Algorithm 1 is consistent.*

**Assumption 3.** $q_-^{(n)} \to +\infty$ *as* $n \to \infty$.

Within Assumption 2 all points have a consistency of 1, and therefore $q = r > 0$ the reliability. Besides, one has $G(x_i^{(n)}, q_i^{(n)}) \leq G(x_{i+1}^{(n)}, q_{i+1}^{(n)})$, i.e., $y_i^{(n)} \leq y_{i+1}^{(n)}$ for all points $i$ and steps $n$. We finally define the sequence of piecewise constant reconstruction functions $F^{(n)}$ as follows.

**Definition 1.** *For each $x \in [a, b]$, we define the reconstructing function $F^{(n)}$ at step $n$ as:*

$$F^{(n)}(x) = \sum_{i=1}^{I^{(n)}-1} y_i^{(n)} \mathbb{1}_{s_i^{(n)}}(x),$$

*and* $F^{(n)}(x_{I^{(n)}}^{(n)}) = F^{(n)}(b) = y_{I^{(n)}}^{(n)}.$

Now let

$$E^+ := \{n \in \mathbb{N};\ \text{WA}^{(n)} \geq \mathcal{E}\}, \qquad E^- := \{n \in \mathbb{N};\ \text{WA}^{(n)} < \mathcal{E}\}, \qquad (12)$$

which are such that $E^+ \cup E^- = \mathbb{N}$ and $E^+ \cap E^- = \varnothing$. In order to prove the convergence (in a sense to be given) of Algorithm 1, we first need to establish the following intermediate results, Proposition 1, Proposition 2, and Proposition 3. They clarify the behaviour of the sequence $\text{WA}^{(n)}$ when points are added to the dataset and the largest area $a_+^{(n)}$ is divided into four parts at each iteration step $n$; see Figure 2.

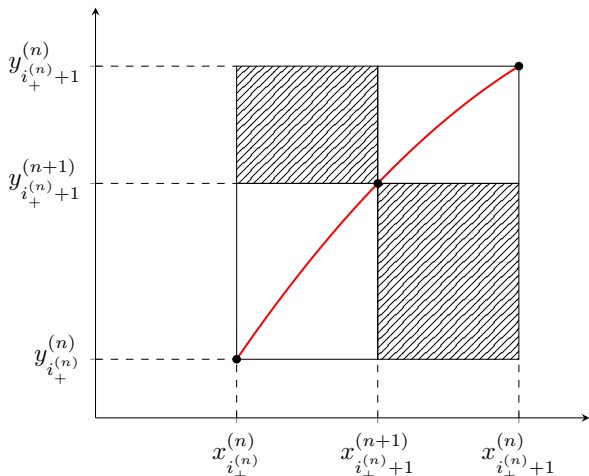

**Figure 2.** New area when one adds a point at the middle of the biggest rectangle.

**Proposition 1.** $E^+$ *is infinite.*

**Proof.** Let us assume that $E^+$ is finite: $\exists N$ such that $\forall n \geq N, n \in E^-$. Therefore we are in the situation $\text{WA}^{(n)} < \mathcal{E}$, the minimum quality $q_-^{(n)}$ of the data goes to infinity, and the total area $A^{(n)}$ is

modified although the evaluation points $\{x_i^{(n)}\}_{i=1}^{I^{(n)}}$ and their number $I^{(n)}$ are unchanged; thus they are independent of $n$. Repeating this step yields

$$\lim_{n\to\infty} A^{(n)} = \sum_{i=1}^{I-1} (x_{i+1} - x_i)(F^\dagger(x_{i+1}) - F^\dagger(x_i)) = A > 0$$

since $F^\dagger$ is monotonically increasing and non-constant on $[a, b]$, and Assumption 1 is used. Consequently $WA^{(n)} \to +\infty$ as $n \to \infty$, that is $WA^{(n)} \geq \mathcal{E} \; \forall n \geq N_1$ for some $N_1$, which is a contradiction. $\square$

The set $E^+$ is therefore of the form

$$E^+ = \bigcup_{k \geq 1} [\![m_k, n_k]\!],$$

where

$$[\![m_k, n_k]\!] := \{n \in \mathbb{N}; \; m_k \leq n \leq n_k\}.$$

Let us introduce the strictly increasing application $\varphi : \mathbb{N} \to \mathbb{N}$ such that $\varphi(p)$ is the $p^{\text{th}}$ element of $E^+$ (in increasing order), and $[\![m_k, n_k]\!] = \varphi([\![p_k + 1, p_{k+1}]\!])$. $p$ is the counter of the elements of $E^+$, and $n$ is the corresponding iteration number.

**Proposition 2.** *Let* $I^{(\varphi(p))} = I^{(\varphi(0))} + p$. *Then*

$$A^{(\varphi(p))} = \sum_{i=1}^{I^{(\varphi(p))}-1} a_i^{(\varphi(p))} = O\left(\frac{1}{\sqrt{p}}\right)$$

*as* $p \to \infty$, *and* $A^{(n)} \to 0$ *as* $n \to 0$.

**Proof.** Let $k \geq 1$ and $n = \varphi(p) \in [\![m_k, n_k]\!]$, where $p \in [\![p_k + 1, p_{k+1}]\!]$. Let $A^{(n)}$ be given by Equation (6), $a_+^{(n)}$ be given y Equation (7), and $i_+^{(n)}$ be given by Equation (8). At iteration $n + 1$ one has:

$$x_i^{(n+1)} = \begin{cases} x_i^{(n)} & \text{for } 1 \leq i \leq i_+^{(n)}, \\ \dfrac{1}{2}\left(x_{i_+^{(n)}}^{(n)} + x_{i_+^{(n)}+1}^{(n)}\right) & \text{for } i = i_+^{(n)} + 1, \\ x_{i-1}^{(n)} & \text{for } i_+^{(n)} + 2 \leq i \leq I^{(n+1)}. \end{cases}$$

Also $y_i^{(n+1)} \leq y_{i+1}^{(n+1)}$ for $1 \leq i \leq I^{(n+1)} - 1$. One may check that $a_+^{(n)} = 2a_{i_+^{(n)}}^{(n+1)} + 2a_{i_+^{(n)}+1}^{(n+1)}$ (see Figure 2) and therefore:

$$A^{(n+1)} = A^{(n)} - a_+^{(n)} + a_{i_+^{(n)}}^{(n+1)} + a_{i_+^{(n)}+1}^{(n+1)} = A^{(n)} - \frac{1}{2}a_+^{(n)}. \tag{13}$$

Besides $A^{(n)} \leq (I^{(n)} - 1)a_+^{(n)}$ so that one has:

$$A^{(n+1)} \leq A^{(n)} - \frac{A^{(n)}}{2(I^{(n)} - 1)}$$

$$\leq A^{(n)}\left(\frac{2(I^{(n)} - 1) - 1}{2(I^{(n)} - 1)}\right),$$

or:

$$A^{(\varphi(p)+1)} \leq A^{(\varphi(p))} \left( \frac{2(I^{(\varphi(p))} - 1) - 1}{2(I^{(\varphi(p))} - 1)} \right). \tag{14}$$

At this stage two situations arise:

- either $p \in [\![ p_k + 1, p_{k+1} - 1 ]\!]$, in which case $\varphi(p) + 1 = \varphi(p+1)$;
- or $p = p_{k+1}$, in which case by our algorithm $A^{(n)}$ is kept constant from $n = n_k + 1$ to $n = m_{k+1}$; that is $A^{(n_k+1)} = A^{(m_{k+1})}$, or:

$$A^{(\varphi(p_{k+1})+1)} = A^{(\varphi(p_{k+1}+1))}.$$

The choice of $k$ being arbitrary, one concludes that Equation (14) also reads $\forall p \in \mathbb{N}$:

$$A^{(\varphi(p+1))} \leq A^{(\varphi(p))} \left( \frac{2(I^{(\varphi(p))} - 1) - 1}{2(I^{(\varphi(p))} - 1)} \right)$$
$$\leq A^{(\varphi(p))} \left( \frac{2(I^{(\varphi(0))} + p - 1) - 1}{2(I^{(\varphi(0))} + p - 1)} \right).$$

Thus:

$$A^{(\varphi(p))} \leq A^{(\varphi(1))} \prod_{i=1}^{p-1} \left( \frac{2(I^{(\varphi(0))} + i - 1) - 1}{2(I^{(\varphi(0))} + i - 1)} \right)$$
$$\leq A^{(\varphi(1))} \prod_{i=1}^{p-1} \left( \frac{1 + \frac{\alpha}{i}}{1 + \frac{\beta}{i}} \right),$$

letting $\alpha = I^{(\varphi(0))} - \frac{3}{2}$ and $\beta = I^{(\varphi(0))} - 1$.

However,

$$\sum_{i=1}^{p} \log \left( 1 + \frac{\alpha}{i} \right) = \alpha \sum_{i=1}^{p} \frac{1}{i} + C_p''$$

where $\lim_{p \to \infty} C_p'' = C''$, and

$$\sum_{i=1}^{p} \frac{1}{i} = \log p + \gamma + \epsilon_p',$$

where $\gamma$ is the Euler constant and $\lim_{p \to \infty} \epsilon_p' = 0$. Consequently:

$$\sum_{i=1}^{p-1} \log \left( 1 + \frac{\alpha}{i} \right) - \sum_{i=1}^{p-1} \log \left( 1 + \frac{\beta}{i} \right) = (\alpha - \beta) \log(p-1) + C_p'$$
$$= (\alpha - \beta) \left[ \log p + \log \left( 1 - \frac{1}{p} \right) \right] + C_p'$$
$$= \log \left( \frac{1}{\sqrt{p}} \right) + C_p,$$

since $\alpha - \beta = -\frac{1}{2}$; again $C_p$ and $C_p'$ are sequences with constant limits $\lim_{p \to \infty} C_p = C$ and $\lim_{p \to \infty} C_p' = C'$. Therefore,

$$\prod_{i=1}^{p-1} \left( \frac{1 + \frac{\alpha}{i}}{1 + \frac{\beta}{i}} \right) = \frac{\mathcal{C}}{\sqrt{p}} (1 + \epsilon_p)$$

where $\mathcal{C}$ is a constant, and $\lim_{p \to \infty} \epsilon_p = 0$. One also concludes that $A^{(n)}$, which is either kept constant or equal to $A^{(\varphi(p))}$, converges to 0 as $n \to \infty$. Hence the claimed results hold. $\square$

**Proposition 3.** *$E^-$ is infinite.*

**Proof.** Let us assume that $E^-$ is finite: $\exists N$ such that $\forall n \geq N, n \in E^+$. Therefore we are in the situation $WA^{(n)} \geq \mathcal{E} > 0$, and $\varphi(n)$ has the form $\varphi(n) = n - n_0, n \geq N$ for some $n_0 \in \mathbb{N}$. From Proposition 2:

$$A^{(n-n_0)} = O\left(\frac{1}{\sqrt{n}}\right),$$

thus $A^{(n)} \to 0$ and $WA^{(n)} \to 0$ as $n \to \infty$ since $q_-^{(n)}$ is kept unchanged, which is a contradiction. $\square$

We now provide three results on the convergence of Algorithm 1. As is to be expected, the algorithm can only be shown to converge uniformly when the target response function $F^\dagger$ is sufficiently smooth; otherwise, the convergence is at best pointwise or in mean.

**Theorem 1** (Algorithm convergence). *Assume that $F^\dagger$ is strictly increasing. Then, for any choice of $\mathcal{E} > 0$, Algorithm 1 is convergent in the following senses:*

- *If $F^\dagger$ is piecewise continuous on $[a, b]$, then $\lim_{n \to \infty} F^{(n)}(x) = F^\dagger(x)$ at all points $x \in [a, b]$ where $F^\dagger$ is continuous;*
- *If $F^\dagger$ is continuous on $[a, b]$, then convergence holds uniformly: $\|F^{(n)} - F^\dagger\|_\infty \xrightarrow[n \to \infty]{} 0$.*

**Proof.** Let $\mathcal{E} > 0$. We know from Propositions 1 and 3 that $WA^{(n)}$ will oscillate about $\mathcal{E}$ in the iterating process as $n \to \infty$, while $\lim_{n \to \infty} q_-^{(n)} = +\infty$ from Assumption 3. Furthermore, let

$$\Delta^{(n)} := \sup_{1 \leq i \leq I^{(n)}-1} \left| x_{i+1}^{(n)} - x_i^{(n)} \right|.$$

Assuming for example that for some $j$, $s_j^{(n)} = [x_j^{(n)}, x_{j+1}^{(n)})$ is never divided in two in the iteration process and is thus independent of $n$, it turns out that $a_j^{(h)} \to (x_{j+1} - x_j)(F^\dagger(x_{j+1}) - F^\dagger(x_j)) > 0$ as $n \to \infty$, which is impossible because $A^{(n)}$ goes to 0 as $n \to \infty$ from Proposition 2. Therefore there exists some $m \in \mathbb{N}^*$ (depending on $n$) such that $\Delta^{(n+m)} \leq \frac{1}{2}\Delta^{(n)}$; also the sequence $\Delta^{(n)}$ is decreasing, hence $\Delta^{(n)} \to 0$ as $n \to \infty$.

Now let $x \in [x_i^{(n)}, x_{i+1}^{(n)})$. Then:

$$\begin{aligned}
\left| F^{(n)}(x) - F^\dagger(x) \right| &= \left| G(x_i^{(n)}, q_i^{(n)}) - F^\dagger(x) \right| \\
&\leq \left| G(x_i^{(n)}, q_i^{(n)}) - F^\dagger(x_i^{(n)}) \right| + \left| F^\dagger(x_i^{(n)}) - F^\dagger(x) \right|.
\end{aligned}$$

However, $x_i^{(n)} \to x$ as $n \to \infty$ because $\Delta^{(n)} \to 0$; thus if $F^\dagger$ is continuous at $x$, the second term on the right hand side above goes to 0 as $n \to \infty$. However, if $F^\dagger$ is continuous everywhere on $[a, b]$, it is in addition uniformly continuous on $[a, b]$ by Heine's theorem, and the second term goes to 0 as $n \to \infty$ uniformly on $[a, b]$. Finally, invoking Assumption 1, the first term on the right hand side above also tends to 0 as $n \to \infty$. This completes the proof. $\square$

**Proposition 4** (Convergence in mean). *Let $F^\dagger : [a, b] \to \mathbb{R}$ be piecewise continuous. Then Algorithm 1 is convergent in mean in the sense that*

$$\|F^{(n)} - F^\dagger\|_1 \xrightarrow[n \to \infty]{} 0.$$

**Proof.** We can check that the sequence $F^{(n)}$ is monotone. Indeed, if $WA^{(n)} < \mathcal{E}$, then by construction we have

$$F^{(n+1)}(x) - F^{(n)}(x) \geq \left( y_{i_-^{(n)}}^{(n+1)} - y_{i_-^{(n)}}^{(n)} \right) \mathbb{1}_{s_-^{(n)}}(x) \geq 0$$

where $s_-^{(n)} = [x_{i_-^{(n)}}^{(n)}, x_{i_-^{(n)}+1}^{(n)})$. However, if $\mathrm{WA}^{(n)} > \mathcal{E}$, then consistency implies that

$$F^{(n+1)}(x) - F^{(n)}(x) \geq \left( y_{i_+^{(n)}+1}^{(n+1)} - y_{i_+^{(n)}}^{(n)} \right) \mathbb{1}_{s_+^{(n+1)}}(x) \geq 0$$

where $s_+^{(n+1)} = [x_{i_+^{(n)}+1}^{(n+1)}, x_{i_+^{(n)}+2}^{(n+1)})$. The claim now follows from the monotone convergence theorem and the fact that $F^{(0)}$ is integrable. □

## 4. Test Cases

To show the effectiveness of Algorithm 1, we try it on two cases, in which $F^{\dagger}$ is a continuous function and a discontinuous function respectively. For both cases, the error between the numerical estimate and the ground truth function is modelled as a random variable following a Log-normal distribution. That is,

$$\forall x \in [a, b], \ \epsilon(x) \sim \mathrm{Log}\mathcal{N}(\mu(x), \sigma^2), \tag{15}$$

with $\sigma^2 = 1$ and $\mu(x)$ is chosen as $\mathbb{P}[0 \leq \epsilon(x) \leq 0.1 \cdot F^{\dagger}(x)] = 0.9$. Thus, the mean $\mu$ is different for each $x \in [a, b]$.

As we have access to the ground truth function and for validation purpose, the quality value associated woth a numerical point is the inverse of the relative error. Moreover, we assume that the initial points are consistent.

For illustrative purposes, we set the parameter $\mathcal{E} = 15$ for the examples considered below.

### 4.1. $F^{\dagger}$ Is a Continuous Function

First, consider the function $F^{\dagger} \in C^0([1, 2], [1, 2])$ defined as follows:

$$F^{\dagger}(x) = \begin{cases} F_1^{\dagger}(x) & \text{if } x \in [1, \frac{3}{2}], \\ F_2^{\dagger}(x) & \text{if } x \in [\frac{3}{2}, 2], \end{cases}$$

with

$$\begin{aligned} F_1^{\dagger}(x) &= a_1 \exp(x^3) + b_1, \\ F_2^{\dagger}(x) &= a_2 \exp((3 - x)^3) + b_2, \end{aligned} \tag{16}$$

where:

$$a_1 = -\frac{1}{2(\exp(1) - \exp(27/8))}, \quad b_1 = \frac{3 - 2\exp(19/8)}{2(1 - \exp(19/8))}, \quad a_2 = -a_1, \quad b_2 = 2a_1 \exp(27/8) + b_1.$$

The target function $F^{\dagger}$ and the reconstructions $F^{(n)}$ obtained through the algorithm for several values of the step $n$ are shown in Figure 3. For each $n$, the reconstruction $F^{(n)}$ is increasing and the initial points are consistent. The $\infty$-norm and 1-norm of the error appear to converge to zero with approximate rates $-0.512$ and $-0.534$ respectively.

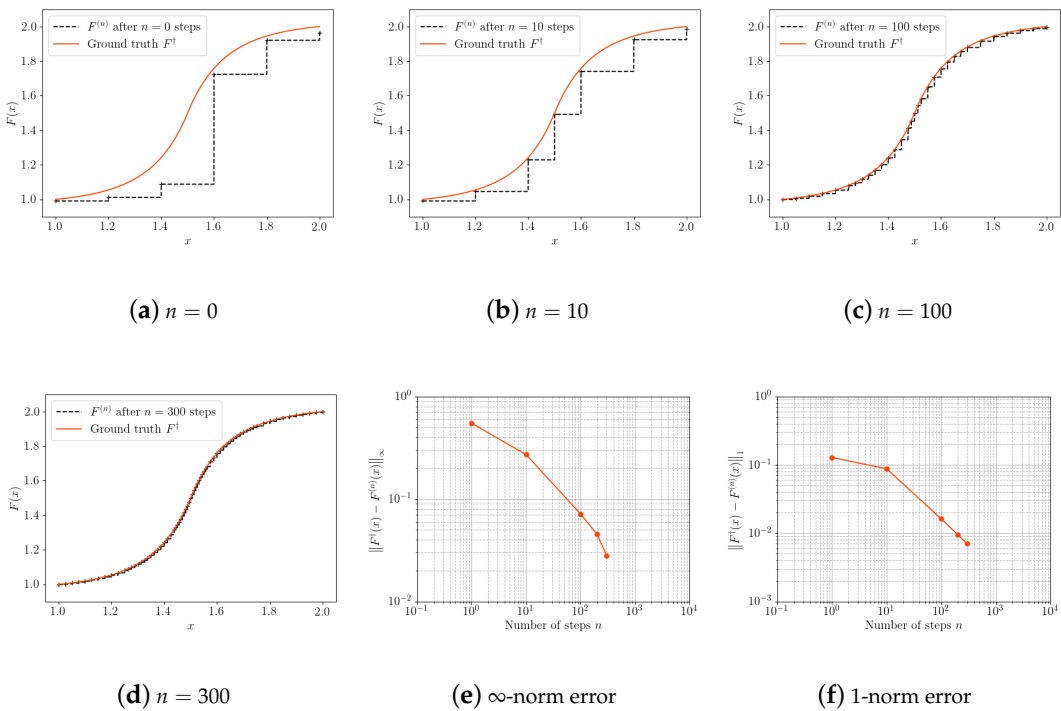

**Figure 3.** Evolution of $F^{(n)}$ and the $\infty$- and 1-norms of the error $F^{\dagger} - F^{(n)}$ as functions of the iteration count, $n$, for a smooth ground truth $F^{\dagger}$.

### 4.2. $F^{\dagger}$ Is a Discontinuous Function

Now, consider the discontinuous function $F^{\dagger}$ defined as follows:

$$F^{\dagger}(x) = \begin{cases} F_1^{\dagger} & \text{if } x \in [1, \frac{3}{2}], \\ F_2^{\dagger} & \text{if } x \in (\frac{3}{2}, 2], \end{cases}$$

where $F_1^{\dagger}$ and $F_2^{\dagger}$ are given by (16), and:

$$a_1 = -\frac{1}{2(\exp(1) - \exp(27/8))}, \qquad b_1 = \frac{3 - 2\exp(19/8)}{2(1 - \exp(19/8))},$$

$$a_2 = \frac{2}{5(\exp(8) - \exp(27/8))}, \qquad b_2 = \frac{10 - 8\exp(37/8)}{5(1 - \exp(37/8))}.$$

Here, $F^{\dagger}$ is piecewise continuous on $[1, \frac{3}{2}]$ and $]\frac{3}{2}, 2]$. In this case, one can apply Proposition 4. The target function $F^{\dagger}$ and the reconstructions $F^{(n)}$ obtained through the algorithm for several values of the step $n$ are shown in Figure 4. Observe that the approximation quality, as measured by the $\infty$-norm of the error $F^{\dagger} - F^{(n)}$, quite rapidly saturates and does not converge to zero. This is to be expected for this discontinuous target $F^{\dagger}$, since closeness of two functions in the supremum norm mandates that they have approximately the same discontinuities in exactly the same places. The 1-norm error, in contrast, appears to converge at the rate $-0.561$.

Regarding computational cost, the number of calls to the numerical model is lower when $F^{\dagger}$ is continuous than when it is discontinuous. For both examples above and for the same number of data points, the number of evaluations of the numerical model (analytical formula in the present case) in the discontinuous case is about six times higher than the number of evaluations in the continuous case. This is because the algorithm typically adds more points near discontinuities and the effort of making them consistent increases the number of calls to the model.

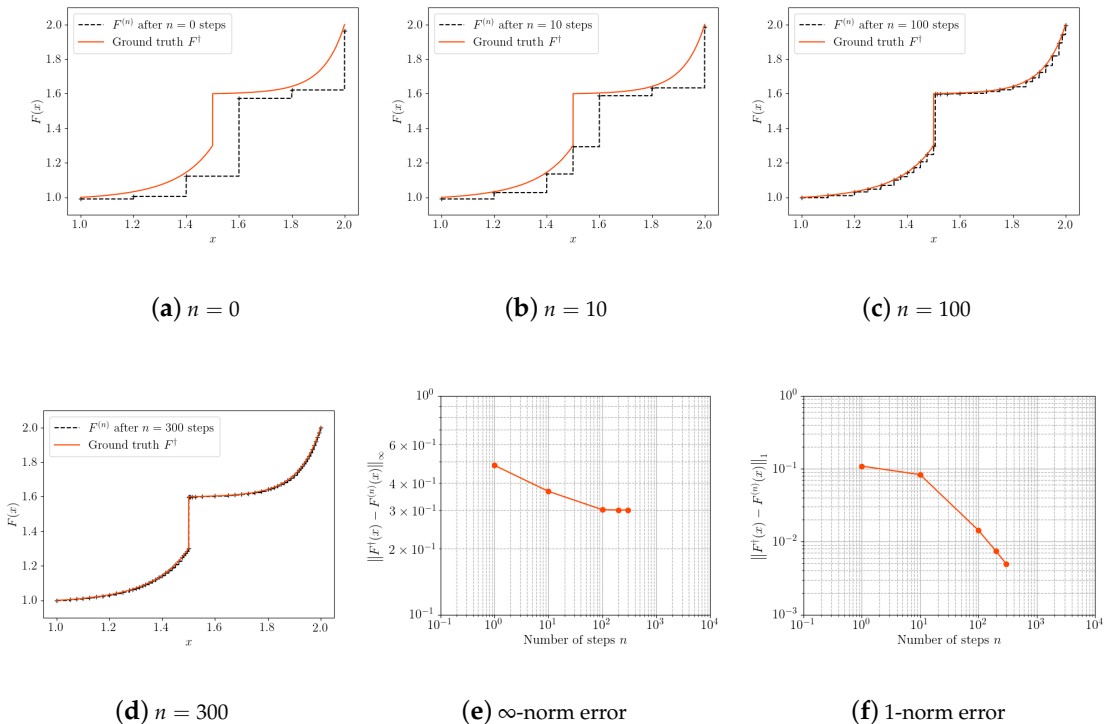

**(a)** $n = 0$            **(b)** $n = 10$            **(c)** $n = 100$

**(d)** $n = 300$          **(e)** $\infty$-norm error      **(f)** 1-norm error

**Figure 4.** Evolution of $F^{(n)}$ and the $\infty$- and 1-norms of the error $F^{\dagger} - F^{(n)}$ as functions of the iteration count, $n$, for a discontinuous ground truth $F^{\dagger}$.

### 4.3. Influence of the User-Defined Parameter $\mathcal{E}$

We consider the case in which $F^{\dagger}$ is discontinuous, as in Section 4.2. We will show the influence of the choice of the parameter $\mathcal{E}$ on the reconstruction function $F^{(n)}$.

#### 4.3.1. Case $\mathcal{E} \ll 1$

Let us consider the case $\mathcal{E} = 10^{-4} \ll 1$. This choice corresponds to the case where one wishes to split over redo the worst quality point. This can be seen on Figure 5 where the worst quality is almost constant over 100 steps while the sum of areas strongly decreases; see Figure 5e and Figure 5f respectively. At each step, the algorithm is adding a new point by splitting the biggest rectangle. One can note on Figure 5f that the minimum of the quality is not constant. It means that when the algorithm added a new data point, the point with the worst quality was not consistent any more and had to be recomputed. In summary, in this case, we obtain more points but with lower quality values.

#### 4.3.2. Case $\mathcal{E} \gg 1$

We now consider the case $\mathcal{E} = 10^{4} \gg 1$. This choice corresponds to the case where one wishes to redo the worst quality point over split. This can be seen on Figure 6 where the sum of areas stays more or less the same over 100 steps while the minimum of the quality surges; see Figure 6f and Figure 6e respectively. There is no new point. The algorithm is only redoing the worst quality point to improve it. To sum up, we obtain fewer points with higher quality values.

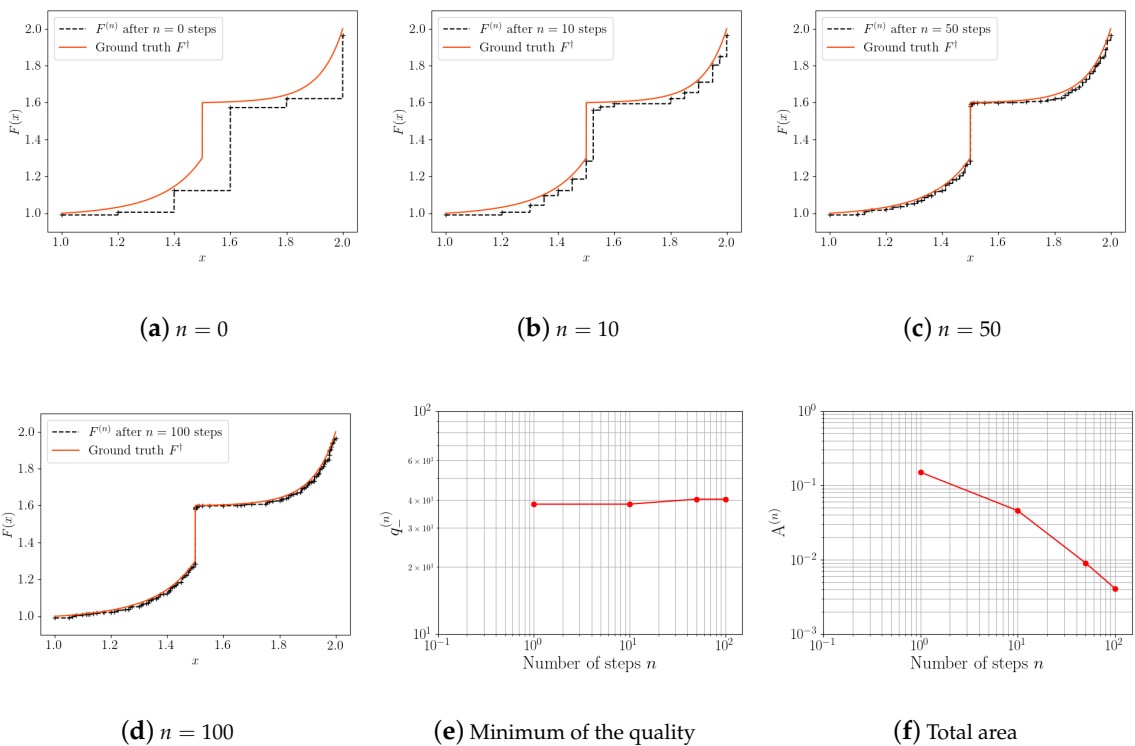

**Figure 5.** Evolution of $F^{(n)}$ and the minimum of the quality and the total area as functions of the iteration count, $n$, for a discontinuous ground truth $F^{\dagger}$ with $\mathcal{E} = 10^{-4}$.

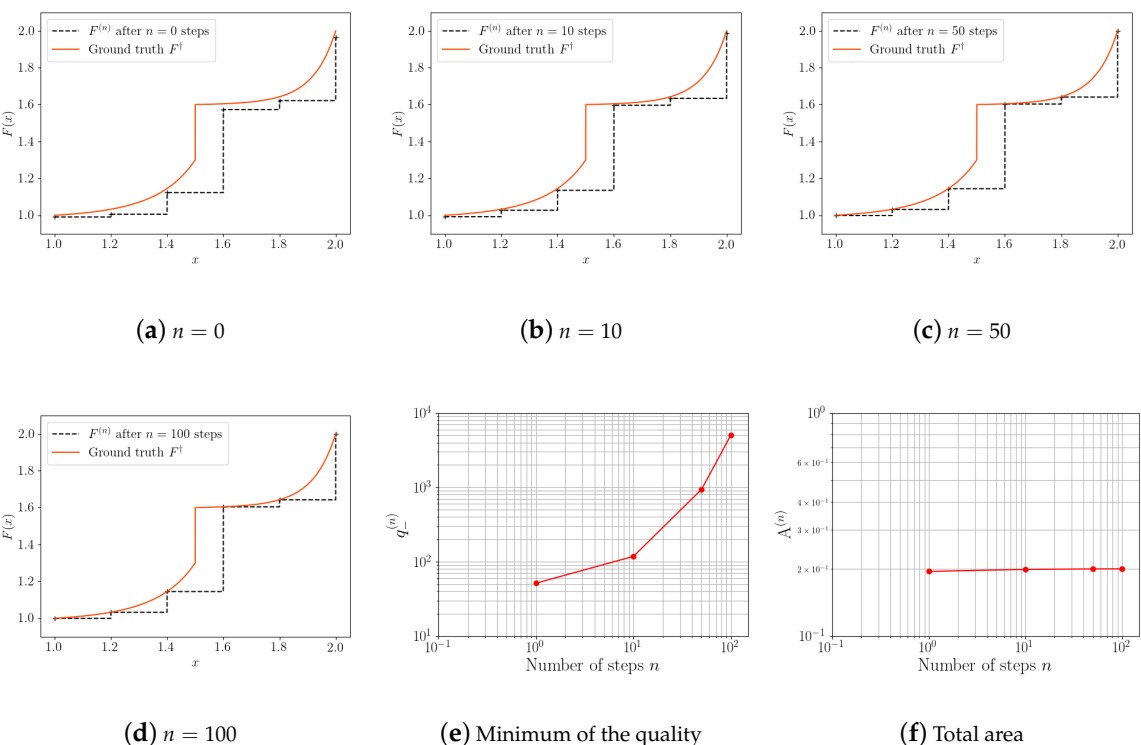

**Figure 6.** Evolution of $F^{(n)}$ and the minimum of the quality and the total area as functions of the iteration count, $n$, for a discontinuous ground truth $F^{\dagger}$ with $\mathcal{E} = 10^{4}$.

## 5. Application to Optimal Uncertainty Quantification

### 5.1. Optimal Uncertainty Quantification

In the *optimal uncertainty quantification* paradigm proposed by Owhadi et al. [7] and further developed by, e.g., Sullivan et al. [8] and Han et al. [9], upper and lower bounds on the performance of an incompletely specified system are calculated via optimisation problems. More concretely, one is interested in the probability that a system, whose output is a function $g^\dagger \colon \mathcal{X} \to \mathbb{R}$ of inputs $\Xi$ distributed according to a probability measure $\mu^\dagger$ on an input space $\mathcal{X}$, satisfies $g^\dagger(\Xi) \le x$, where $x$ is a specified performance threshold value. We emphasise that although we focus on a scalar performance measure, the input $\Xi$ may be a multivariate random variable.

In practice, $\mu^\dagger$ and $g^\dagger$ are not known exactly; rather, it is known only that $(\mu^\dagger, g^\dagger) \in \mathcal{A}$ for some admissible subset $\mathcal{A}$ of the product space of all probability measures on $\mathcal{X}$ with the set of all real-valued functions on $\mathcal{X}$. Thus, one is interested in

$$\underline{P}_{\mathcal{A}}(x) := \inf_{(\mu,g)\in\mathcal{A}} \mathbb{P}_{\Xi\sim\mu}[g(\Xi) \le x] \quad \text{and} \quad \overline{P}_{\mathcal{A}}(x) := \sup_{(\mu,g)\in\mathcal{A}} \mathbb{P}_{\Xi\sim\mu}[g(\Xi) \le x].$$

The inequality

$$0 \le \underline{P}_{\mathcal{A}}(x) \le \mathbb{P}_{\Xi\sim\mu^\dagger}[g^\dagger(\Xi) \le x] \le \overline{P}_{\mathcal{A}}(x) \le 1$$

is, by definition, the tightest possible bound on the quantity of interest $\mathbb{P}_{\Xi\sim\mu^\dagger}[g^\dagger(\Xi) \le x]$ that is compatible with the information used to specify $\mathcal{A}$. Thus, the optimal UQ perspective enriches the principles of worst- and best-case design to account for distributional and functional uncertainty. We concentrate our attention hereafter, without loss of generality, on the least upper bound $\overline{P}_{\mathcal{A}}(x)$.

**Remark 1.** *The main focus of this paper is the dependency of $\overline{P}_{\mathcal{A}}(x)$ on $x$. In practice, an underlying task is, for any individual $x$, reducing the calculation of $\overline{P}_{\mathcal{A}}(x)$ to a tractable finite-dimensional optimisation problem. Central enabling results here are the* reduction theorems *of (Owhadi et al. [7], Section 4), which loosely speaking, say that if, for each $g$, $\{\mu \mid (\mu, g) \in \mathcal{A}\}$ is specified by a system of $m$ equality or inequality constraints on expected values of arbitrary test functions under $\mu$, then for the determination of $\overline{P}_{\mathcal{A}}(x)$ it is sufficient to consider only distributions $\mu$ that are convex combinations of at most $m + 1$ point masses; the optimisation variables are then the $m$ independent weights and $m + 1$ locations in $\mathcal{X}$ of these point masses. If $\mu$ factors as a product of distributions (i.e., $\Xi$ is a vector with independent components), then this reduction theorem applies componentwise.*

As a function of the performance threshold $x$, $\overline{P}_{\mathcal{A}}(x)$ is an increasing function, and so it is potentially advantageous to determine $\overline{P}_{\mathcal{A}}(x)$ jointly for a wide range of $x$ values using the algorithm developed above. Indeed, determining $\overline{P}_{\mathcal{A}}(x)$ for many values of $x$, rather than just one value, is desirable for multiple reasons:

1. Since numerical optimisation to determine $\overline{P}_{\mathcal{A}}(x)$ may be affected by errors, computing several values of $\overline{P}_{\mathcal{A}}(x)$ could lead to validate their consistency as the function $x \mapsto \overline{P}_{\mathcal{A}}(x)$ must be increasing;
2. The function $\overline{P}_{\mathcal{A}}(x)$ can be discontinuous. Thus, by computing several values of $\overline{P}_{\mathcal{A}}(x)$, one can highlight potential discontinuities and can identify key threshold values of $x \mapsto \overline{P}_{\mathcal{A}}(x)$.

### 5.2. Test Case

For the application of Algorithm 1 to OUQ, we study the robust shape optimization of the two-dimensional RAE2822 airfoil [10] (Appendix A6) using ONERA's CFD software *elsA* [11]. The following example is taken from Dumont et al. [12]. The shape of the original RAE2822 is altered using four bumps located at four different locations: 5%, 20%, 40%, and 60% of the way along the chord $c$ (see Figure 7). These bumps are characterised by B-splines functions.

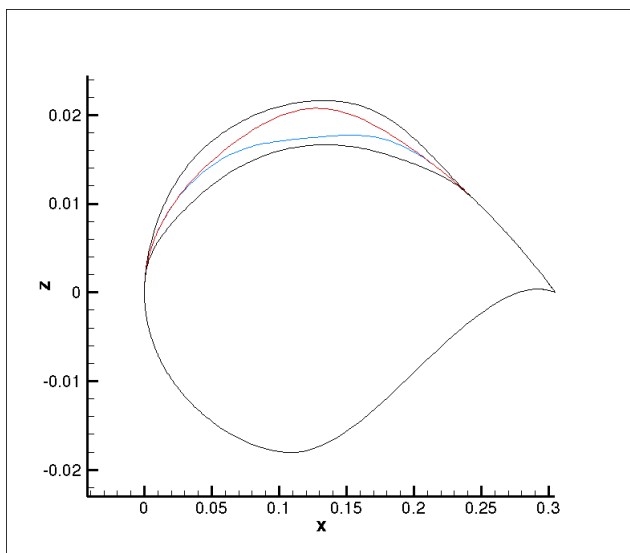

**Figure 7.** Black lines: Maximum and minimum deformation of the RAE2822 profile. Red: Maximum deformation of the third bump alone. Blue: Minimum deformation of the third bump alone. This image is taken from Dumont et al. [12].

The lift-to-drag ratio $\frac{C_l}{C_d}$ of the RAE2822 wing profile (see Figure 8) at Reynolds Number $Re = 6.5 \times 10^6$, Mach number $M_\infty = 0.729$ and angle of attack $\alpha = 2.31°$ is chosen as the performance function $g^\dagger$ with inputs $\Xi = (\Xi_1, \Xi_2, \Xi_3, \Xi_4)$, where $(\Xi_i)_{i=1...4}$ is the amplitude of each bump. They will be considered as random variables over their respective range given in Table 1.

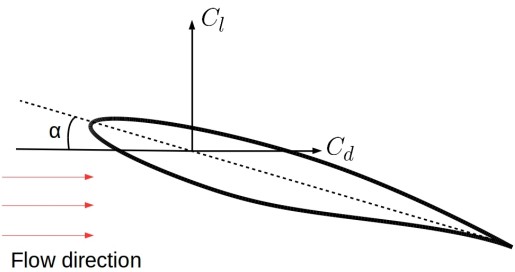

**Figure 8.** Picture depicting the lift $C_l$ and the drag $C_d$ of an airfoil.

**Table 1.** Range of each input parameter.

|  | **Range** | **Law** |
|---|---|---|
| Bump 1: $\Xi_1$ | [−0.0025c; +0.0025c] | $\mu_1^\dagger$: Beta law with $\alpha = 6, \beta = 6$ |
| Bump 2: $\Xi_2$ | [−0.0025c; +0.0025c] | $\mu_2^\dagger$: Beta law with $\alpha = 2, \beta = 2$ |
| Bump 3: $\Xi_3$ | [−0.0025c; +0.0025c] | $\mu_3^\dagger$: Beta law with $\alpha = 2, \beta = 2$ |
| Bump 4: $\Xi_4$ | [−0.0025c; +0.0025c] | $\mu_4^\dagger$: Beta law with $\alpha = 2, \beta = 2$ |

The corresponding flow values are the ones described in test case #6 together with the wall interferences corrections formulas given in [13] (Chapter 6) and in [14] (Section 5.1). Moreover, we will assume that $(\Xi_i)_{i=1...4}$ are mutually independent. An ordinary Kriging procedure has been chosen to build a metamodel (or response surface) of $g^\dagger$, which is identified with the actual response function $g^\dagger$ in the subsequent analysis. A tensorised grid of 9 equidistributed abscissas for each parameter is used. The model is then based on $N = 9^4 = 6561$ observations. In that respect, a Gaussian kernel

$$K(\Xi, \Xi') = \exp\left(-\frac{1}{2}\sum_{i=1}^{4}\frac{(\Xi_i - \Xi_i')^2}{\gamma_i^2}\right)$$

has been chosen, where $\Xi = (\Xi_1, \Xi_2, \Xi_3, \Xi_4)$ and $\Xi' = (\Xi_1', \Xi_2', \Xi_3', \Xi_4')$ are inputs of the function $g^\dagger$, and where $\gamma = (\gamma_1, \gamma_2, \gamma_3, \gamma_4)$ are the parameters of the kernel. These parameters are chosen to minimize the variance between the ground truth data defined by the $N$ observations and their Kriging metamodel $g^\dagger$. The responce surfaces in the $(\Xi_1, \Xi_3)$ plan for two values of $(\Xi_2, \Xi_4)$ are shown in Figure 9.

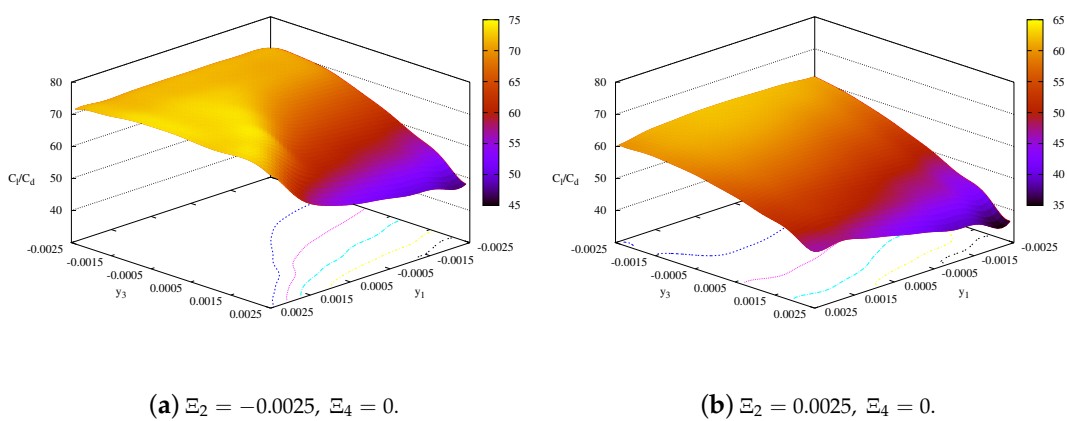

**(a)** $\Xi_2 = -0.0025$, $\Xi_4 = 0$.        **(b)** $\Xi_2 = 0.0025$, $\Xi_4 = 0$.

**Figure 9.** Response surface in the $(\Xi_1, \Xi_3)$ plane with $(\Xi_2 = -0.0025, \Xi_4 = 0)$ **(a)** and $(\Xi_2 = 0.0025, \Xi_4 = 0)$ **(b)**. These images are taken from Dumont et al. [12].

One seeks to determine $\overline{P}_{\mathcal{A}}(x) := \sup_{\mu \in \mathcal{A}} \mathbb{P}_{\Xi \sim \mu}[g^\dagger(\Xi) \le x]$, where the admissible set $\mathcal{A}$ is defined as follows:

$$\mathcal{A} = \left\{ (g, \mu) \,\middle|\, \begin{array}{c} \Xi \in \mathcal{X} = \mathcal{X}_1 \times \mathcal{X}_2 \times \mathcal{X}_3 \times \mathcal{X}_4 \\ g : \mathcal{X} \mapsto \mathcal{Y} \text{ is known equal to } g^\dagger \\ \mu = \mu_1 \otimes \mu_2 \otimes \mu_3 \otimes \mu_4 \\ \mathbb{E}_{\Xi \sim \mu}[g(\Xi)] = \text{LD} \end{array} \right\}. \tag{17}$$

A priori, finding $\overline{P}_{\mathcal{A}}(x)$ is not computationally tractable because it requires a search over a infinite-dimensional space of probability measures defined by $\mathcal{A}$. Nevertheless, as described briefly in Remark 1, it has been shown in Owhadi et al. [7] that this optimisation problem can be reduced to a finite-dimensional one, where now the probability measures are products of finite convex combinations of Dirac masses.

**Remark 2.** *The ground truth law $\mu^\dagger$ of each input variable given in Table 1 is only used to compute the expected value $\mathbb{E}_{\Xi \sim \mu}[g(\Xi)] = \text{LD}$. This expected value is computed with $10^4$ samples.*

**Remark 3.** *The admissible set $\mathcal{A}$ from (17) can be understood as follows:*

- *One knows the range of each input parameter $(\Xi_i)_{i=1,\dots,4}$;*
- *$g$ is exactly known as $g = g^\dagger$;*
- *$(\Xi_i)_{i=1,\dots,4}$ are independent;*
- *One only knows the expected value of $g$: $\mathbb{E}_{\Xi \sim \mu}[g(\Xi)]$.*

The optimisation problem of determining $\overline{P}_{\mathcal{A}}(x)$ for each chosen $x$ was solved using the Differential Evolution algorithm of Storn and Price [15] within the *mystic* optimisation framework [16].

Ten iterations of Algorithm 1 have been performed using $\mathcal{E} = 1 \times 10^4$. The evolution of $\overline{P}_{\mathcal{A}}(x)$ as function of the iteration count, $n$, is shown in Figure 10. At $n = 0$—see Figure 10a—two consistent points are present at $x = 57.51$ and $x = 67.51$. At this step, $\text{WA}^{(0)} = 35289$. As $\text{WA}^{(0)} \geq \mathcal{E}$, at next step $n = 1$, the algorithm adds a new point at the middle of the biggest rectangle—see Figures 10b and 11b. After $n = 10$ steps, eight points are now present in total with a minimum quality increasing from 5000 to 11,667 and with a total area decreasing from 7.05 to 0.84; see Figure 11a and Figure 11b respectively.

The number of iterations in this complex numerical experiment has been limited to 10 because obtaining new or improved data points consistent throughout the optimization algorithm may take up to two days (wall-clock time on a personal computer equipped with an Intel Core i5-6300HQ processor with 4 cores and 6 MB cache memory) for one single point. This running time is increased further for data points of higher quality. Nevertheless, this experiment shows that the proposed algorithm can be used for real-world examples in an industrial context.

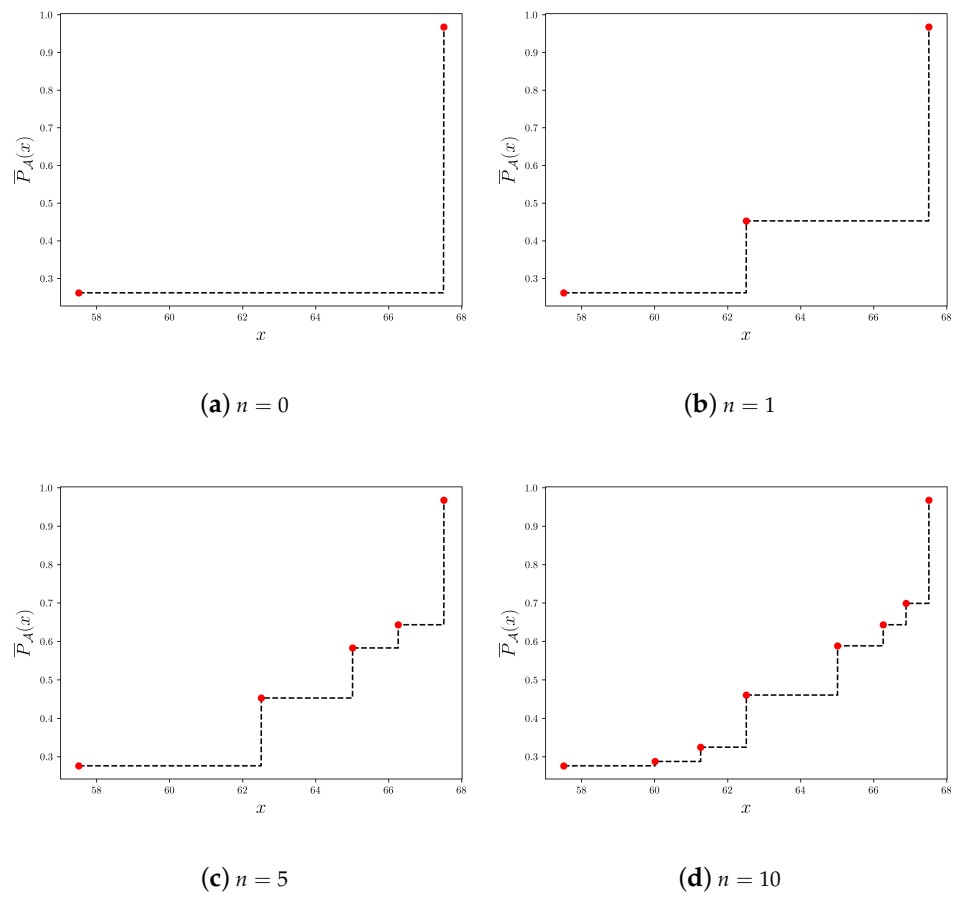

(**a**) $n = 0$      (**b**) $n = 1$

(**c**) $n = 5$      (**d**) $n = 10$

**Figure 10.** Evolution of $\overline{P}_{\mathcal{A}}(x)$ as function of the iteration count, $n$.

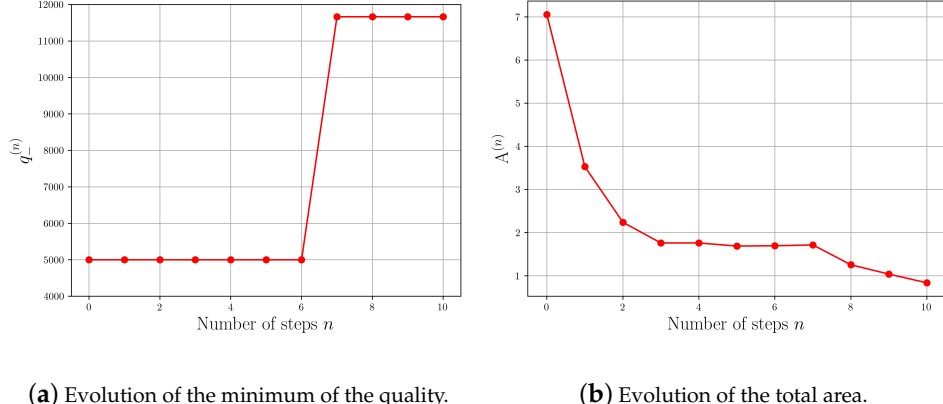

(**a**) Evolution of the minimum of the quality.

(**b**) Evolution of the total area.

**Figure 11.** Evolution of the minimum of the quality and the total area as function of the iteration count, *n*.

## 6. Concluding Remarks

In this paper we have developed an algorithm to reconstruct a monotonically increasing function such as the cumulative distribution function of a real-valued random variable, or the least upper bound of the performance criterion of a system as a function of its performance threshold. In particular, this latter setting has relevance to the optimal uncertainty quantification (OUQ) framework of [7] we have in mind for applications to real-world incompletely specified systems. The algorithm uses imperfect pointwise evaluations of the target function, subject to partially controllable one-sided errors, to direct further evaluations either at new sites in the function's domain or to improve the quality of evaluations at already-evaluated sites. It allows for some flexibility at targeting either strategy through a user-defined "exchange rate" parameter, yielding an approximation of the target function with a few high-quality points or alternatively more lower-quality points. We have studied its convergence properties and have applied it to several examples: known target functions that are either continuous and discontinuous, and a performance function for aerodynamic design of a well-documented standard profile in the OUQ setting.

Algorithm 1 is reminiscent of the classical PAVA approach to isotonic regression that applies to statistical inference with order restrictions. Examples of its use can be found in shape constrained or parametric density problems as illustrated in e.g., [6]. Possible improvements and extensions of our algorithm include weighting the areas $a_i^{(n)}$ as they are summed up to form the total weighted area $\mathrm{WA}^{(n)}$ driving the iterative process, in order to optimally enforce both the addition of "steps" $s_i^{(n)}$ in the reconstruction function $F^{(n)}$ of Definition 1, and the improvement of their "heights" $y_i^{(n)}$. This could be achieved considering for example the following alternative definition $i_+^{(n)} = \arg\max_i \{(I^{(n)} - i - 1)a_i^{(n)}\}$ in Algorithm 1, which results in both adding a step to the $i_+^{(n)}$-th current one and possibly improving all subsequent evaluations $y_i^{(n+1)}$, $i > i_+^{(n)}$. We may further envisage to adapt the ideas elaborated in this research to the reconstruction of convex functions by extending the notion of consistency. These perspectives shall be considered in future works.

**Author Contributions:** Conceptualization, L.B. and T.J.S.; methodology, L.B. and T.J.S.; software, L.B.; validation, J.-L.A., É.S., and T.J.S.; formal analysis, L.B., J.-L.A., É.S., and T.J.S.; investigation, L.B.; resources, L.B., J.-L.A., É.S., and T.J.S.; data curation, L.B.; writing–original draft preparation, L.B.; writing—review and editing, L.B., J.-L.A., É.S., and T.J.S.; visualization, L.B.; supervision, É.S. and T.J.S.; project administration, T.J.S.; funding acquisition, L.B., É.S., and T.J.S. All authors have read and agreed to the published version of the manuscript.

**Funding:** The work of J.-L.A. and É.S. has been partially supported by ONERA within the Laboratoire de Mathématiques Appliquées pour l'Aéronautique et Spatial (LMA$^2$S). L.B. is supported by a CDSN grant from the French Ministry of Higher Education (MESRI) and a grant from the German Academic Exchange Service (DAAD), Program #57442045. T.J.S. has been partially supported by the Freie Universität Berlin within the Excellence Strategy of the DFG, including project TrU-2 of the Excellence Cluster "MATH+ The Berlin Mathematics Research Center" (EXC-2046/1, project 390685689) and DFG project 415980428.

**Conflicts of Interest:** The authors declare no conflict of interest. The funders had no role in the design of the study; in the collection, analyses, or interpretation of data; in the writing of the manuscript, or in the decision to publish the results.

## Abbreviations

The following abbreviations are used in this manuscript:

CFD     Computational Fluid Dynamics
DOAJ   Directory of open access journals
MDPI   Multidisciplinary Digital Publishing Institute
OUQ    Optimal Uncertainty Quantification
PAVA   Pool-Adjacent-Violators Algorithm

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
