# Peer review of "Adaptive Reconstruction of Imperfectly Observed Monotone Functions, with Applications to Uncertainty Quantification"

_algorithms, doi:10.3390/a13080196_

Round 1
Reviewer 1 Report
Report:
In this paper Authors present an adaptive algorithm for reconstruction of a monotonic function. The algorithm is clearly described and illustrated with some figures. Morover, authors analyse the convergence of the algorithm and its performance against some benchmark functions. The authors proposal to use imperfect pointwise evaluations of the target function seems interesting. Overall, the paper is well written and I recommend publication in Algorithms.
Author Response
Please see the attachment for reply to comments by the second referee.

Reviewer 2 Report
Report

Round 2
Reviewer 2 Report
The authors have carefully revised the manuscript following the suggestions from the report.
The manuscript is now suitable for publication, in my opinion.